# FLOWING FROM OBSERVED TO FUTURE FRAMES FOR EFFICIENT VIDEO PREDICTION

## ABSTRACT

This paper introduces a novel methodology for fast and memory-efficient video prediction. Our method, dubbed FlowFrames, fine-tunes a pre-trained text-to-video flow model to learn a vector field between the observed and future frame distributions. Two design choices are key. First, we introduce inherent optimal couplings, utilizing consecutive video chunks during training as a practical proxy for optimal couplings, which results in straighter flows. Second, we incorporate target inversion, injecting the inverted latent of the target chunk into the input representation to strengthen correspondences and improve visual fidelity. By flowing directly from observed to future frames, instead of the common combination of input frames with noise to generate future frames, we reduce the dimensionality of the model input by a factor of two. The proposed method, fine-tuned from LTXV and Wan, surpasses the state-of-the-art scores across quantitative evaluations with FID and FVD, with as few as five neural function evaluations. We will release the code and models of our method to the public.

## 1 INTRODUCTION

Recent advances in image and video generation have dramatically improved the quality of synthetic media, largely due to diffusion and flow-based models. Building on this progress, researchers have turned to *video prediction*: synthesizing a plausible sequence of future frames given a preceding segment of video, which may consist of real or previously generated frames. In essence, the goal of video prediction is to generate a temporally coherent and semantically consistent continuation of the input video segment. Success in video prediction would enable key applications in autonomous driving, human motion forecasting, and immersive AR/VR environments.

Yet video prediction remains difficult. As the future is inherently uncertain, models must satisfy two sometimes-competing needs: *temporal coherence*, to ensure realistic motion and smooth dynamics, and *visual fidelity*, to preserve sharp details and semantic consistency across frames. To tackle this, state-of-the-art approaches condition on past frames and use Gaussian noise to generate future ones (Gao et al., 2024; Hassan et al., 2024; HaCohen et al., 2024; Yin et al., 2025). While effective, this design has key drawbacks: *high memory cost* since the model must process both the conditioning frames and noise of output dimensionality, and *slow inference* as multiple neural function evaluations are required. This is even more of a problem considering that video prediction applications such as autonomous driving and AR/VR require the methods to work on embedded machines, in almost-real-time. In this work, we introduce **FlowFrames**, a simple but powerful alternative on the video prediction task. Our model receives a chunk of video frames as input and generates the future video chunk as a plausible continuation of the given input chunk, keeps sematic coherence with the latter, and propagates global contextual information.

FlowFrames fine-tunes a pre-trained text-to-video model (e.g. LTXV (HaCohen et al., 2024) or Wan (Wan et al., 2025)) by modifying the initial distribution from multivariate normal distribution to the distribution of input video frames. Then it learns a mapping between the input and future frame distributions by flow matching (Lipman et al., 2023; Liu et al., 2022; Albergo & Vanden-Eijnden, 2023). By *flowing directly from observed to future frames*, FlowFrames bypasses the conventional paradigm of conditioning the model on input frames and using Gaussian noise for the generation of future frames. This results in a twofold reduction of model input dimensionality. Additionally, our model is trained using inherent optimal couplings, where input and future video chunks from

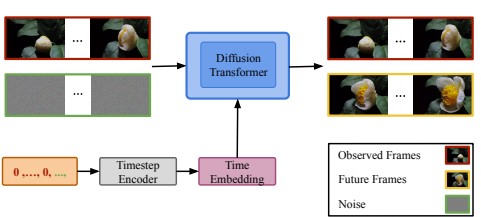 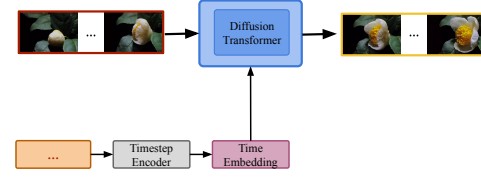

(a) Conventional input design like LTXVCondition (HaCohen et al., 2024)

(b) Proposed input design

Figure 1: (a) In conventional approaches, both the observed frames and additional Gaussian noise of the same size as the future chunk are provided to the model at inference. Zero timesteps are used for the conditioning frames. (b) **FlowFrames** removes the noise; the model directly learns a vector field from observed to future frames and thus enables sampling from solely an observed video chunk. This simple design halves the input size. Also, by leveraging inherent optimal couplings and target inversion at training, our approach exceeds the baselines quantitatively with as few as five neural function evaluations. Although visualized in pixel space, the methods operate in latent space.

the same sequence serve as the source and target, respectively. This approach leads to straighter flow trajectories and consequently reduces the number of neural function evaluations required at inference to outperform prior methods quantitatively. FlowFrames also uses inverted latents of the target video chunk, achieving improved visual quality. Our approach is capable of generating coherent video continuations without explicit temporal conditioning on the given input video chunk. Fig. 1 illustrates the key difference between the input design adopted by conventional methods and FlowFrames.

We summarize the primary contributions of this paper as follows:

- We propose a simple yet novel approach to video prediction by directly flowing from observed to future frames, leveraging $2\times$ more memory-efficient input design, which yields - on the LTXV backbone as an example - approximately $50\%$ lower peak GPU memory relative to LTXVCondition (HaCohen et al., 2024).

- Our method leverages inherent optimal couplings of input and future frames as a proxy for the true optimal couplings, reducing the required number of neural function evaluations (NFEs) for state-of-the-art quantitative results (e.g., 4× fewer NFEs on the LTXV backbone). In addition, our method leverages target inversion for enhanced visual fidelity.

- We conduct comprehensive experiments and ablations on OpenVid (Nan et al., 2024) and NuScenes (Caesar et al., 2020) datasets, demonstrating that our method outperforms state-of-the-art world and flow-based auto-regressive text-to-video models in quantitative scores with significant reduction in GPU memory usage and NFEs.

## 2 RELATED WORK

**Text-to-Video Models.** Text-to-video (T2V) diffusion (Wang et al., 2023a;b; Chen et al., 2023; Wu et al., 2023; Blattmann et al., 2023; Khachatryan et al., 2023; Ma et al., 2025b; Chen et al., 2024; Yang et al., 2025) and flow models (HaCohen et al., 2024; Wan et al., 2025; Zheng et al., 2024; Kong et al., 2025; Peng et al., 2025; Ma et al., 2025a; Polyak et al., 2025; Chen et al., 2025a; Jin et al., 2025) are a class of conditional generative models that synthesize videos from text prompts. Stable Video Diffusion (SVD) (Blattmann et al., 2023) focuses on data curation and extends Stable Diffusion (SD) (Rombach et al., 2022) by temporal layers to capture inter-frame dynamics. SVD faces challenges with scalability, long-term temporal modeling and T2V alignment. To overcome these, CogVideoX (Yang et al., 2025) employs a 3D variational autoencoder (VAE) (Kingma & Welling, 2022) for spatio-temporal compression and a diffusion transformer (DiT) (Peebles & Xie, 2023) with full 3D attention, where text and visual tokens are jointly processed via self-attention (Vaswani et al., 2023) to enhance semantic alignment. CogVideoX is computationally expensive due to the compression rate of its VAE. In contrast, HaCohen et al. (2024) apply a four times higher

VAE compression rate than CogVideoX and pretrain a 2B DiT via flow matching (Esser et al., 2024; Lipman et al., 2023; Albergo & Vanden-Eijnden, 2023). Wan (Wan et al., 2025) adopts similar architectural and training principles – DiT, 3D VAE, flow matching – offering scalable models (1.3B and 14B DiTs) for T2V synthesis.

**World and Flow-Based Autoregressive T2V Models for Video Prediction.** Recent video prediction methods are part of world (Gao et al., 2024; Hassan et al., 2024) and flow-based auto-regressive T2V models (HaCohen et al., 2024; Yin et al., 2025). On the one hand, Vista (Gao et al., 2024) augments SVD with a latent-replacement scheme that injects up to three historical frames, and introduces dynamics-enhancement and structure-preservation losses to improve realism. However, Vista struggles with computational efficiency and controlability. To address the latter, GEM predicts future frames from a reference frame with control over DINOv2 (Oquab et al., 2023) features, human poses, and ego-trajectories. GEM adopts progressive noise levels on frames in training for future frame generation. On the other hand, CausVid (Yin et al., 2025) distills a bi-directional T2V model into a uni-directional autoregressive student model using asymmetric distribution matching distillation. It employs a block-wise causal attention along with per-block noise levels, preserving bi-directional attention within local blocks while enforcing causality across them. Finally, LTXV-Condition (HaCohen et al., 2024) is pre-trained using different amounts of noise levels per-token allowing generation of a video sequence from a conditioning chunk which is given to the model as input with zero timestep. While all of the above-mentioned methods support prediction of future frames given an input chunk, we provide a fundamentally different viewpoint on video prediction by directly flowing from observed to future frames.

# 3 METHODOLOGY

This section begins with an overview of flow matching and flow matching (Lipman et al., 2023; Liu et al., 2022; Albergo & Vanden-Eijnden, 2023) with optimal couplings (Pooladian et al., 2023; Tong et al., 2024). It then outlines FlowFrames, which directly trains a vector field between observed and future frames with inherent optimal couplings and integrates target inversion into the pipeline.

## 3.1 PRELIMINARIES

**Flow Matching:** Flow matching (Lipman et al., 2023; Liu et al., 2022; Albergo & Vanden-Eijnden, 2023) is a training paradigm in generative modeling. It learns to transform a sample from an initial distribution $p_{init}(x_0)$ – typically the standard multivariate normal distribution $x_0 \sim \mathcal{N}(0, I_d)$ – into $x_1$ where $x_1$ comes from the data distribution $p_{data}$ and $d$ refers to the dimensionality of the data. This is accomplished by first perturbing the target data sample during the training at a given time step $t$ via a convex combination with noise:

$$x = tz + (1-t)x_0, \tag{1}$$

where $t \sim \mathcal{U}_{[0,1]}$ is sampled uniformly from the unit interval. The perturbed input $x$ is then passed to a time-conditioned neural network $u_t^\theta(x)$. The model is trained to predict the velocity vector $(z - x_0)$, using the conditional flow matching loss:

$$\mathcal{L}_{\text{CFM}}(\theta) = ||u_t^\theta(x) - (x_1 - x_0)||^2. \tag{2}$$

Once $u_t^\theta(x)$ has been trained, a new sample from $p_{data}$ can be generated by first sampling $x_0$ from the standard multivariate normal distribution and then following the velocity field predicted by the model using Euler discretization. In the case of linear probability paths, the initial distribution is not restricted to being Gaussian; a flow matching model can be trained between two arbitrary distributions. This motivates us to consider training a flow matching model between distributions of observed and future frames with the goal to generate future frames given an input video chunk.

**Flow Matching With Optimal Couplings:** Multiple strategies exist for sampling $x_0$ and $x_1$ at training (e.g. independent sampling (Lipman et al., 2023; Tong et al., 2024) or the use of optimal couplings (OC) (Pooladian et al., 2023; Kornilov et al., 2024)). Formally, given the joint distribu-

---

**Algorithm 1** Flowing From Observed To Future Frames

---

1: **Require:** pretrained $u_t^{\theta^*}$, $\Pi(p_{\text{input\_frames}}, p_{\text{future\_frames}})$, $\rho$
2: $u_t^\theta \leftarrow u_t^{\theta^*}$
3: **for** $x_0, x_1 \sim \Pi$ **with** $(x_0, x_1)$ **inherent optimal couplings do**
4: $\quad \mu_1, \sigma_1 = \text{VAE}(x_1), \quad x_1 \leftarrow \mu_1$
5: $\quad \mu_0, \sigma_0 = \text{VAE}(x_0)$
6: $\quad \hat{x}_1 = \text{RF-Solver-Inversion}(u_t^{\theta^*}, x_1)$ (Wang et al., 2025) $\qquad \triangleright$ Compute inverted latent
7: $\quad$ **if** $p < \rho$ **then** $x_0 \leftarrow \mu_0 + \sigma_0 \hat{x}_1$ **else** $x_0 \leftarrow \mu_0$ $\qquad \triangleright$ Apply Target Inversion
8: $\quad t \sim \mathcal{U}[0, 1]$
9: $\quad x \leftarrow (1 - t)x_0 + t x_1$
10: $\quad \mathcal{L}_{\text{CFM}}(\theta) = \left\| u_t^\theta(x) - (x_1 - x_0) \right\|^2$
11: $\quad$ Update $\theta$ using GD on $\mathcal{L}_{\text{CFM}}(\theta)$
12: **end for**

---

tion $\Pi(p_{init}, p_{data})$, the optimal coupling is given by the minimizer of the following optimization problem:

$$\min_{\pi \in \Pi(p_{\text{init}}, p_{\text{data}})} \left( \mathbb{E}_{\pi(x_0, x_1)} \left[ \|x_0 - x_1\|^2 \right] \right) \tag{3}$$

This formulation corresponds to minimizing the squared 2-Wasserstein distance. Employing optimal couplings for sampling pairs of noise and data during training facilitates the learning of straighter flows for $u_t^\theta(x)$, thereby reducing the number of sampling steps required at inference time to achieve high-quality results in comparison to sampling independently during the training. However, computing optimal couplings in high-dimensional spaces such as images or videos remains computationally intractable in practice. To address this, Pooladian et al. (2023); Tong et al. (2024) propose approximating optimal couplings within a batch at training, demonstrating that such an approach leads to straighter trajectories and a reduced number of inference steps in contrast to non-optimal couplings.

### 3.2 FLOWING FROM OBSERVED TO FUTURE FRAMES

Given observed frames, the goal of video prediction is to generate the next video chunk that is semantically coherent with the given ones (Ming et al., 2025). We propose to directly flow from input to the future frames using flow matching. Our motivation is twofold. (1) By flowing directly from observed to future frames, FlowFrames avoids the conventional approach – such as LTXVCondition (HaCohen et al., 2024) and CausVid (Yin et al., 2025) – of providing both observed frames and additional noise at inference, thereby halving the input dimensionality. (2) We hypothesize that learning a vector field between observed and future frames may facilitate faster convergence and enable more efficient inference-time sampling compared with mapping from noise to future frames. Formally, given $x_0$ from the distribution of input frames $p_{\text{input\_frames}}$ and $x_1$ from the distribution of future frames $p_{\text{future\_frames}}$ our goal is to learn a parametric vector field $u_t^\theta(x)$ to predict the velocity between input and future frames. In practice, flow matching is performed in the latent space of a pre-trained 3D VAE (Kingma & Welling, 2022). Algorithm 1 presents our training algorithm.

**Inherent Optimal Couplings as Input and Future Frames.** As discussed in Section 3.1, learning a vector field with optimal couplings leads to improved convergence and straighter sample trajectories. However, in high-dimensional settings such as videos, computing exact optimal couplings under a cost function is computationally intractable. Thus, we propose a practical and structure-aware approximation: treating temporally adjacent video chunks of the same video as inherently optimal or near-optimal couplings. Specifically, we partition a given video into consecutive segments: the first segment serves as the input chunk $x_0$, and the second as the future chunk $x_1$. This pairing leverages the natural temporal continuity and semantic coherence within a single video, which makes it significantly more consistent than random pairings of chunks across different videos. Crucially, such intra-video couplings incur no additional computational overhead, yet act as a strong proxy for true optimal transport couplings. Empirically, we observe that using these temporally ordered pairs at training result in decreased loss values (Fig. 4 (left)) and reduce the number of neural function evaluations required for high-quality inference in comparison to independent pairs (Fig. 6a).

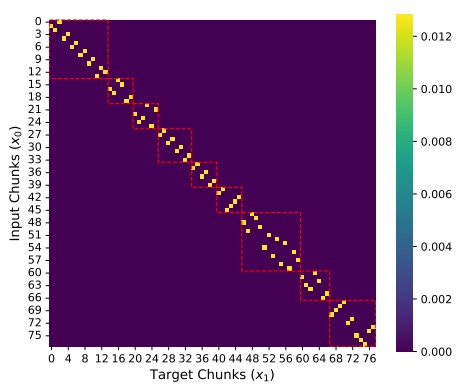 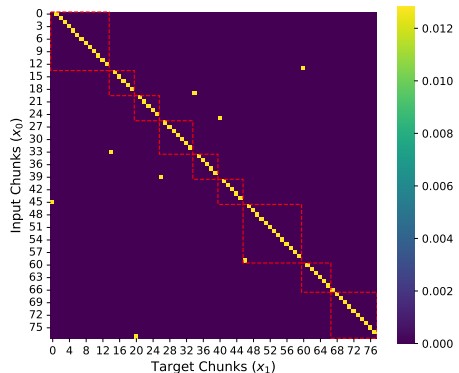

(a) Disallowing a video chunk from matching with itself.

(b) Matching only with future chunks from the same video.

Figure 2: Optimal Transport (OT) plan heatmaps between video chunks. We compute pairwise OT plans between a batch of video chunks, solving the Monge–Kantorovich OT problem with uniform marginals. Red dashed lines mark individual video boundaries. The structure of the transport plans reveals strong alignment between temporally adjacent chunks from the same video, even without explicit temporal constraints in (b).

To empirically validate our hypothesis that consecutive chunks from the same video approximate optimal couplings, we compute pairwise Optimal Transport (OT) plans between a batch of video chunks. Specifically, we randomly sample ten videos from our training set, OpenVid (Nan et al., 2024), and divide each into chunks of 41 frames. We then solve the Monge–Kantorovich OT problem (Kantorovich, 1948) subject to uniform marginals:

$$\min_{\pi \in \mathbb{R}^{n \times n}} \sum_{i=1}^{n} \sum_{j=1}^{n} M_{ij}\, \pi_{ij} \quad \text{s.t.} \quad \sum_{j=1}^{n} \pi_{ij} = \tfrac{1}{n}, \ \ \forall i, \quad \sum_{i=1}^{n} \pi_{ij} = \tfrac{1}{n}, \ \ \forall j, \tag{4}$$

where $\pi_{ij}$ indicates the probability of matching source $i$ with target $j$, $n$ refers to the number of video chunks and $M_{ij}$ denotes the pairwise squared Euclidean cost between the $i$-th and $j$-th video chunks. Fig. 2 presents heatmaps of the resulting optimal transport plans under two different masking strategies: in Fig. 2a, self-matching is disallowed (i.e. a video chunk can not match with itself), whereas in Fig. 2b, each chunk is permitted to match only with future chunks from the same video. Red dashed lines in Fig. 2 demarcate the boundaries between individual videos. The spurious matches observed for the final chunk of each video in Fig. 2b are natural artifacts arising from the absence of valid future chunks. The overall coupling patterns exhibit a strong preference for temporally adjacent segments within the same video; even in the absence of explicit temporal constraints; suggesting that such chunk pairs can serve as structure-aware, computation-free approximations of optimal couplings.

**Target Inversion.** Replacing the multivariate normal distribution with the empirical distribution of observed frames when fine-tuning the given pre-trained T2V model, also changes the generative objective faced by the pre-trained vector field, since the target vector field becomes the difference between future and observed frames. To bridge this mismatch, we introduce Target Inversion (TI): given a chunk of future frames $x_1$, we recover an inverted latent $\hat{x}_1$ using RF-Solver inversion (Wang et al., 2025) and the given pre-trained model such that the Euler solver reconstructs $x_1$ from $\hat{x}_1$. Then, during training we, with probability $\rho$, leverage $\hat{x}_1$ when sampling a latent representation for the input frames. We hypothesize that this strategy biases the model toward learning the residual shift-and-scale transformation applied to the inverted target latent. While we do not observe measurable gains in convergence speed or standard quantitative metrics (Fig. 4), we consistently see improved perceptual fidelity in predicted videos (Fig. 5).

Table 1: GPU memory usage scales linearly with effective volume $V$: $\mathrm{Mem}(V) \approx k \cdot (V/10^6) + b$. Here $k$ and $b$ are the OLS slope and intercept; smaller $k$ means lower GPU memory usage. We achieve a twofold reduction in $k$. Best $k$ values in each backbone group are shaded in gray (i.e. among methods with the same backbone that we fine-tuned on).

| Method | Backbone | $k$ (MB / $10^6$) | $b$ (MB) |
|---|---|---|---|
| Vista | SVD(1.5B) | 20244.37 | 18430.21 |
| GEM | | 6934.35 | 4194.67 |
| LTXVCondition | LTXV(2B) | 7103.22 | -660.92 |
| Ours | | 3552.32 | -308.35 |
| CausVid | Wan(1.3B) | 93.35 | 4003.58 |
| Ours | | 48.64 | 3998.74 |

## 4 EXPERIMENTS

### 4.1 EXPERIMENTAL SETTINGS

**Datasets:** We train our models on a random subset of 400K videos from OpenVid (Nan et al., 2024). To mitigate abrupt scene transitions in individual videos, we apply a histogram-based scene change detector to segment videos into coherent clips. For evaluation, we use randomly sampled set of 2000 videos from OpenVid and another random sample of 2000 videos from the NuScenes (Caesar et al., 2020) validation set used by Vista (Gao et al., 2024).

**Baselines and Evaluation Metrics:** FlowFrames is compared against recent world models, including Vista (Gao et al., 2024) and GEM (Hassan et al., 2024), as well as autoregressive T2V models, LTXVCondition (HaCohen et al., 2024) and CausVid (Yin et al., 2025). We analyze GPU memory overhead during generation. The video prediction quality is evaluated using the Fréchet Inception Distance (FID)(Heusel et al., 2018) and the Fréchet Video Distance (FVD)(Unterthiner et al., 2019) against the ground truth future chunk. For evaluation, we generate video predictions given an observed video chunk. Owing to input-frame constraints in baselines such as Vista and the NuScenes validation set, we use 17 input frames and predict the subsequent 17 frames. For ablation studies on the method and NFE, we set the number of input and future frames to 41 and use $256 \times 384$. More details on the evaluation protocol are provided in the Appendix A.

**Implementation Details.** We fine-tune our models initialized from LTXV (2B) and Wan (1.3B) on H100 GPUs with a total batch size of 14,336 for 1450 steps, using AdamW (Loshchilov & Hutter, 2019) with a learning rate of $2 \times 10^{-4}$. We use 17 and 41 frame video chunks at training, spatially resized to $256 \times 384$ when using LTXV as a backbone, and to $240 \times 416$ when using Wan as a backbone. All training hyperparameters are provided in Appendix A.

### 4.2 MAIN RESULTS

**Analysis of GPU Memory Overhead.** Table 1 presents an analysis of GPU memory overhead during generation in comparison to the baseline. We define the effective volume $V$ as the product of the latent input tensor dimensions, $c \times f \times h \times w$ where $c$ is the number of channels, $f$ the number of frames, and $h$ and $w$ the spatial resolution. For each method, we evaluate the average end-to-end GPU overhead (including VAE encoding and decoding) over 100 runs at multiple effective volumes. Because different methods employ different VAE compression ratios, we ensure that the effective volumes are matched across models for a fair comparison, see Appendix A for details. To characterize scaling behavior, we assume a linear relationship of the form $\mathrm{Mem}(V) \approx k \cdot (V/10^6) + b$ where $k$ denotes the slope (memory increase per million effective volume units) and $b$ the intercept (constant overhead). $k$ and $b$ are estimated using ordinary least squares (OLS) regression and reported in Table 1. We compare our LTXV-based model to LTXVCondition and our Wan-based model to CausVid. In both cases, FlowFrames achieves almost a twofold reduction in $k$, consistent with our design motivation of reducing input dimensionality.

Table 2: Quantitative comparison on OpenVid and NuScenes validation sets. Our LTXV-based model outperforms LTXVCondition with four times fewer total NFEs, while our Wan-based model surpasses CausVid with similar NFEs and roughly half the input dimensionality. ↓ indicates lower is better. The best and second-best scores are shaded dark and light gray, respectively, within each backbone family (i.e. among methods sharing the same backbone from which we fine-tuned).

| Method | Backbone | Total NFE | OpenVid | | NuScenes | |
|---|---|---|---|---|---|---|
| | | | FID ↓ | FVD ↓ | FID ↓ | FVD ↓ |
| Vista | SVD(1.5B) | 50 | 0.377 | 184.327 | 3.497 | 368.082 |
| GEM | | 117 | 1.470 | 413.600 | 1.885 | 251.133 |
| LTXVCondition | LTXV(2B) | 40 | 0.459 | 134.125 | 1.749 | 228.721 |
| Ours | | 5 | 0.482 | 129.199 | 1.132 | 185.479 |
| | | 10 | 0.439 | 124.967 | 1.079 | 172.861 |
| CausVid | Wan(1.3B) | 5 | 0.299 | 379.157 | 1.277 | 812.457 |
| Ours | | 5 | 0.100 | 105.512 | 0.452 | 158.970 |
| | | 10 | 0.063 | 98.455 | 0.331 | 125.632 |

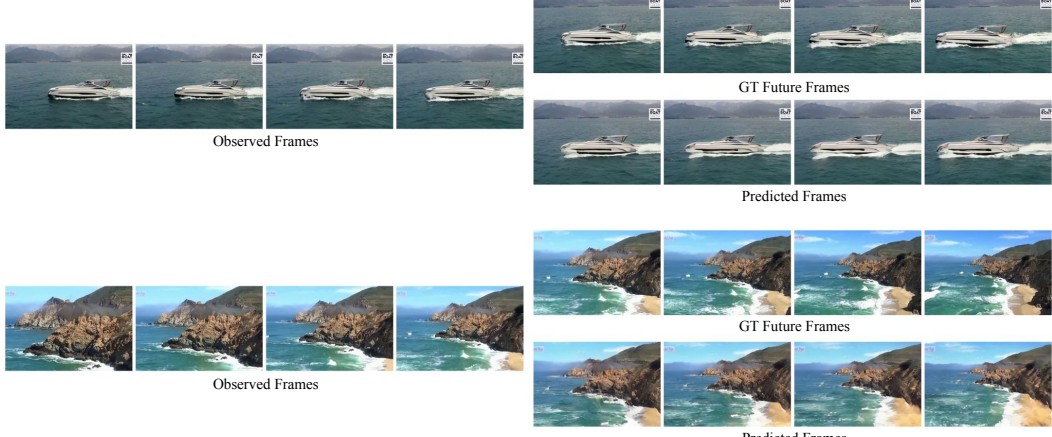

Figure 3: Qualitative results on the OpenVid validation set. Our LTXV-based FlowFrames produces temporally coherent and physically plausible video continuations. Frames are visualized with a stride of 13. GT = ground truth. For example, notice how the boat moves from right to left, a motion that continues in the ground-truth future frames and is also generated by our method. Similarly, observe the gradual camera zoom-out in the sand-and-rock scene, which our method also successfully predicts. More examples are provided in Appendix C with videos in the supplementary material.

**Sampling Efficiency and Quantitative Results.** Table 2 provides a quantitative comparison between baseline methods and FlowFrames (fine-tuned from LTXV and Wan) on the OpenVid and NuScenes validation sets and the total number of neural function evaluations (NFE) per method. The total NFE reflects the number of network passes through the diffusion (or flow) backbone. For CausVid, NFE depends on the size of the KV cache, which we set equal to the number of input frames (i.e. 17). Our LTXV-based model achieves superior FID and FVD scores compared to LTXVCondition, while requiring four times fewer NFEs. Likewise, our Wan-based model outperforms CausVid at similar NFEs and having half the input model dimensionality. Moreover, our Wan-based model attains these results using a straightforward flow matching loss, without additional distillation procedures as required by CausVid.

**Qualitative Results.** Fig. 3 presents visualizations of video prediction results on the OpenVid validation set, generated by our LTXV-based model with total NFE of 10 at a resolution of $256 \times 384$ with 41 observed and future frames. The predicted videos exhibit detailed motion, maintain temporal

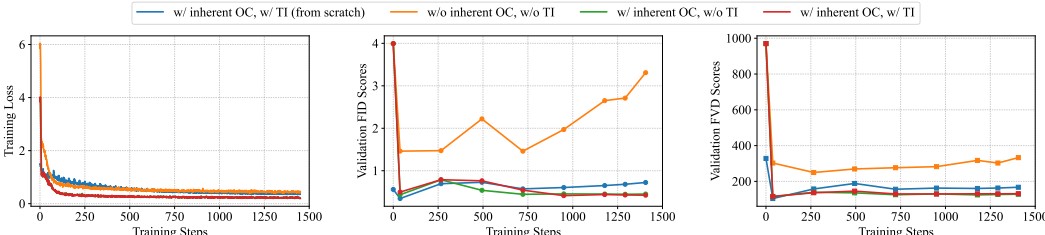

Figure 4: Training loss (left), validation FID (middle), and FVD (right) across four experimental set-ups. Training from scratch with inherent Optimal Couplings (OC) and Target Inversion (TI) is ineffective in terms of training loss. Fine-tuning without inherent OC+TI yields worse FID and FVD than all other settings. Fine-tuning with inherent OC+TI attains the lowest loss and best validation FID and FVD across configurations; relative to inherent OC-only fine-tuning, quantitative scores are comparable, but inherent OC+TI yields better visual quality.

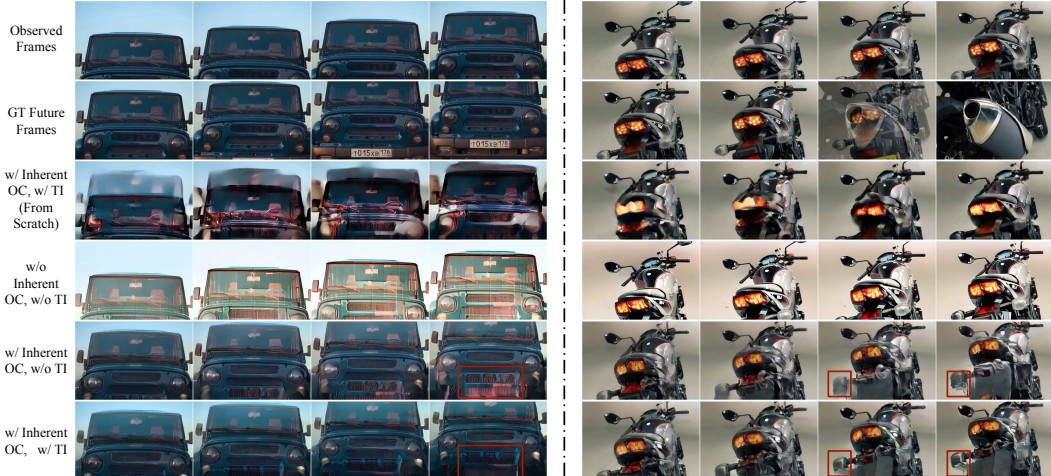

Figure 5: Visual comparison across four settings. Frames are shown with a stride of 13. GT refers to ground truth. Training from scratch w/ OC+TI shows visual artifacts, indicating poor convergence. Fine-tuning w/o OC+TI yields temporal incoherence and color shifts. Fine-tuning w/ OC+TI produces sharper, more consistent details than OC-only fine-tuning (see red rectangles). Overall, w/ OC+TI setting achieves the best visual quality. More results are in Appendix C.

coherence without explicit conditioning on the input frames, and yield logically and physically plausible continuations of the given input video. Additional visual results are provided in Appendix C.

### 4.3 ABLATION STUDIES

**Effect of Fine-tuning, Inherent OC and TI.** Fig. 4 ablates the principal design choices of FlowFrames: fine-tuning from a pre-trained T2V flow model (LTXV) rather than training from scratch; using inherent optimal couplings (OC) as a proxy for the true optimal couplings at training; and employing target inversion (TI). We compare training losses and validation metrics (FID, FVD) across four configurations: (1) training from scratch with inherent OC+TI, (2) fine-tuning without inherent OC and without TI, (3) fine-tuning with inherent OC but without TI, and (4) fine-tuning with inherent OC+TI (Algorithm 1). In training from scratch we initialize the DiT of LTXV with random weights. The implementation details of (2) and (3) are provided in Appendix B. Two main observations emerge. First, training from scratch with inherent OC+TI yields higher losses than fine-tuning from LTXV with inherent OC+TI, underscoring the importance of leveraging pre-trained model's prior knowledge about video generation for video prediction. Second, omitting inherent OC and TI at fine-tuning increases losses and degrades FID and FVD relative to fine-tuning with inherent OC+TI. Although fine-tuning without OC+TI shows losses similar to training from

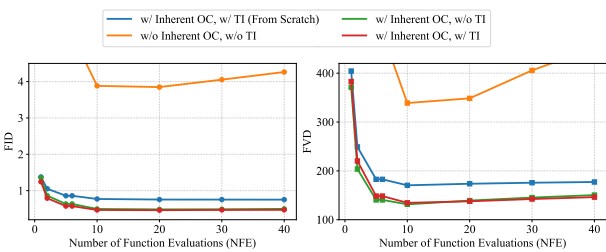 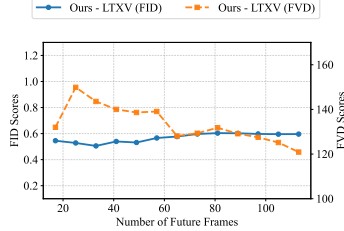

(a) Ablation on the number of function evaluations (NFE).  (b) Ablation on number of frames.

Figure 6: Ablations on NFE and number of frames: (a) With inherent OC+TI, 5–10 NFEs equate or surpass 40 NFEs on FID/FVD and outperform fine-tuning without inherent OC/TI, indicating FlowFrames preserves high visual and temporal quality with fewer NFEs; (b) FlowFrames keeps FID/FVD stable as number of future frames grows, generalizing beyond its 17/41-frame training.

scratch with inherent OC+TI, qualitative inspection reveals future frames that are inconsistent with the conditioning frames (Fig. 5); moreover, continuing fine-tuning with inherent OC+TI beyond step 38 further improves validation FID. Finally, fine-tuning with inherent OC+TI and with inherent OC-only attains comparable losses and FID and FVD, yet inherent OC+TI yields visibly higher perceptual quality in the predicted videos (Fig. 5).

**Ablation on NFE.** Fig. 6a reports a quantitative ablation over NFEs for FlowFrames fine-tuned from LTXV across the four experimental settings indicated above (see Appendix C for visual results). Using 5–10 NFEs match or outperform the 40-NFE setting on both FID and FVD when fine-tuning with inherent OC and TI, demonstrating that high perceptual quality and temporal coherence can be achieved with notably fewer sampling steps. Excluding inherent OC and TI results in substantially worse FID and FVD at NFE = 40 than fine-tuning with inherent OC and TI at NFE = 10. This supports our design choice that leveraging inherent optimal couplings at training enables efficient inference without sacrificing fidelity.

**Ablation on Long Video Prediction.** Fig. 6b presents a quantitative analysis of performance as a function of the number of predicted frames with visual results in Appendix C. FlowFrames, fine-tuned from LTXV, maintains consistent FID and FVD as the number of future frames grows – despite being trained only on 17- and 41-frame video chunks – indicating stable per-frame fidelity, temporal coherence, and effective generalization to long video prediction across the evaluated ranges.

## 5 CONCLUSION

This paper introduced FlowFrames, a next video chunk prediction approach that directly flows from input to future frames. Our method fine-tunes text-to-video priors (e.g., LTXV/Wan) from the distribution of observed to future frames, adopting inherent optimal couplings via consecutive chunks, and integrating target inversion into the training pipeline. Empirically, FlowFrames achieves state-of-the-art FID and FVD scores with as few as five neural function evaluations, while delivering $2\times$ reduction in model input dimensionality relative to baseline methods.

**Limitation and Future Work.** Our training experiments use fixed-length chunks (17 and 41 frames). The method empirically generalizes to longer video prediction but does not yet model variable-length inputs. Future work includes training with heterogeneous chunk lengths to better match future-duration distributions and adding controllability via explicit conditioning (e.g., rendering maps, camera trajectories) for user-directed dynamics. Further limitations and directions are in Appendix D.

**Reproducibility Statement.** Training, experimental, and ablation details are in Appendices A and B. Training and inference codes are in the supplementary material and will be released publicly.

**Large Language Model Use.** Large language models (e.g., ChatGPT-5 (OpenAI, 2025)) were used solely for grammar and proofreading and had no role in this work's conception, analysis, or results.

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

# APPENDIX

## A  IMPLEMENTATION AND EVALUATION DETAILS

**Training Details.** The proposed FlowFrames method is fine-tuned from two backbones: LTXV (HaCohen et al., 2024) and Wan (Wan et al., 2025). All ablation studies are run exclusively on LTXV as LTXV's VAE (Kingma & Welling, 2022) provides a higher overall compression ratio than Wan, yielding faster training and lower compute/memory cost under the same hardware budget (i.e. for the Wan backbone, we double the number of devices relative to LTXV to ensure a matched batch size.). Unless stated otherwise, we use the AdamW optimizer (Loshchilov & Hutter, 2019) with a learning rate of 0.9. The complete set of hyperparameters used for fine-tuning from both LTXV and Wan are listed in Table 3.

Table 3: Fine-tuning hyper-parameters used in our experiments.

| Configuration | LTXV-based | Wan-based |
|---|---|---|
| Batch Size / GPU | 64 | 32 |
| Accumulate Step | 8 | 8 |
| Optimizer | AdamW | AdamW |
| $\beta_1$ | 0.9 | 0.9 |
| $\beta_2$ | 0.99 | 0.99 |
| Learning Rate | 0.0002 | 0.0002 |
| Learning Rate Schedule | Linear | Cosine |
| Training Steps | 1450 | 1450 |
| Resolution | 256×384 | 240×416 |
| Number of Frames | 17, 41 | 17, 41 |
| Shifting | True | True |
| Weighting Scheme | Logit Normal | Uniform |
| Num Layers | 28 | 30 |
| $p$ | 0.7 | 0.7 |
| Pre-trained Model | LTX-Video-2b-v0.9.5 | Wan2.1-T2V-1.3B |
| Sampler | FlowMatchEulerDiscreteScheduler (Esser et al., 2024) | UniPCMultistepScheduler (Zhao et al., 2023) |
| Sample Steps | 40 | 50 |
| Guide Scale | 3.5 | 5 |
| Device | NVIDIA H100 80 GB ×28 | NVIDIA H100 80 GB ×56 |
| Training Strategy | AMP / DDP / BFloat16 | AMP / DDP / BFloat16 |

**Evaluation Setup.** We evaluate FlowFrames against world (Gao et al., 2024; Hassan et al., 2024) and autoregressive HaCohen et al. (2024); Yin et al. (2025) text-to-video baselines in a video prediction setting with a fixed number of frames: each model receives 17 conditioning (or input) frames and predicts the next 17 frames. The 17/17 choice is driven by the maximum sequence length that fits in memory for Vista (Gao et al., 2024) (i.e. 34 total frames per sample) on a single NVIDIA H100 GPU with 80 GB of memory. To ensure a fair comparison, we adopt the same 17-frame input and 17-frame output for all methods, including ours. Table 4 reports all evaluation hyperparameters across methods (e.g., input/generation resolution, classifier-free guidance scale (Ho & Salimans, 2022), and other runtime settings).

**NuScenes Protocol.** For NuScenes dataset (Caesar et al., 2020), we use the validation split from Vista (150 scenes; 750 videos per camera position). We aggregate three camera views – FRONT, BACK, and FRONT-LEFT – and randomly sample 2,000 videos for evaluation to match the sample count of our OpenVid (Nan et al., 2024) validation set. The exact validation indices for OpenVid and NuScenes are provided in the supplementary material.

**Details on Effective Volume for GPU Memory Analysis.** We compare GPU memory usage across world and autoregressive text-to-video models that rely on different backbones and therefore different VAE compression ratios. To ensure a fair comparison, we match (or closely approximate)

Table 4: Evaluation hyper-parameters across all methods being compared. We use the following abbreviations for this table: H = Height, W = Width, In = the number of conditioning or input frames, Out = the number of output frames, Out-L = the number of output frames in the latent space, Blk = frames per block, NFE = number of function evaluations, CFG = classifier-free guidance scale, Rnd = number of sampling rounds.

| Method | H | W | In | Out (img/lat) | Out-L | Blk (frames) | Total NFE | CFG | Rnd |
|---|---|---|---|---|---|---|---|---|---|
| Vista | 576 | 1024 | 17 | 17 | 17 | - | 50 | 2.5 | 1 |
| GEM | 576 | 576 | 17 | 17 | 17 | - | 117 | 1.5 | 1 |
| CausVid | 480 | 832 | 17 | 17 | 5 | 5 | 5 | 1 | - |
| Self-Forcing | 480 | 832 | 17 | 17 | 5 | 5 | 6 | 1 | - |
| LTXVCondition | 256 | 256 | 17 | 17 | 3 | - | 40 | 3.5 | - |
| Ours (LTXV) | 256 | 384 | 17 | 17 | 3 | - | 5/6/10 | 3.5 | - |
| Ours (Wan) | 240 | 416 | 17 | 17 | 5 | - | 5/6/10 | 5 | - |

Table 5: Representative data used in the GPU–memory analysis across five effective volumes. We use the following abbreviations in this table: $V_{\text{SVD}}$ = the effective volume for methods based on the SVD backbone, $V_{\text{LTXV/Wan}}$ = the effective volume for methods that use LTXV or Wan as a backbone, $F_{\text{SVD}}$ = the latent number of conditioning frames used in methods based on the SVD backbone, $F_{\text{LTXV}}$ = the latent number of conditioning (or input) frames used in methods based on the LTXV backbone and $F_{\text{Wan}}$ = the latent number of conditioning (or input) frames used in methods that use the Wan backbone.

| # | $V_{\text{SVD}}$ | $V_{\text{LTXV/Wan}}$ | $F_{\text{SVD}}$ | $F_{\text{LTXV}}$ | $F_{\text{Wan}}$ |
|---|---|---|---|---|---|
| 1 | 294,912 | 299,520 | 5 | 41 | 9 |
| 2 | 368,640 | 399,360 | 8 | 57 | 13 |
| 3 | 479,232 | 499,200 | 10 | 73 | 17 |
| 4 | 589,824 | 599,040 | 13 | 89 | 21 |
| 5 | 884,736 | 898,560 | 16 | 137 | 33 |

the effective volume per run and per method. Vista and GEM (Hassan et al., 2024) use the SVD (Blattmann et al., 2023) backbone; LTXVCondition uses LTXV (HaCohen et al., 2024); CausVid (Yin et al., 2025) uses Wan (Wan et al., 2025); our method is fine-tuned and evaluated with LTXV and Wan backbones. We characterize the VAE latent space by $c \times f \times h \times w$ (the number of channels, the number of frames, height and width): SVD $4 \times 1 \times 8 \times 8$, LTXV $128 \times 8 \times 32 \times 32$, Wan $16 \times 4 \times 8 \times 8$. Table 5 lists representative configurations used in the main-text GPU memory study across five effective volumes, including effective volumes for SVD, LTXV, and Wan, and the latent number of conditioning (or input) frames for each run. For consistency, SVD-based methods are run at $576 \times 1024$ resolution, and LTXV/Wan-based methods at $480 \times 832$.

## B  ABLATION ALGORITHMS FOR CORE DESIGN CHOICES

In addition to the main training algorithm described in the paper, in this section we include two auxiliary variants used in our ablation studies for the assessment of the primary design decisions of FlowFrames. Specifically, we provide pseudocode for: fine-tuning without Inherent Optimal Couplings (OC) and without Target Inversion (Algorithm 2); and fine-tuning with Inherent OC but without Target Inversion (Algorithm 3). In Algorithm 2, observed and future frames are sampled independently from their respective distributions, with no target inversion applied. By contrast, Algorithm 3 incorporates Inherent OC, ensuring coupling between observed and future frames as described in the main text, while still omitting target inversion.

---

**Algorithm 2** Flowing From Observed To Future Frames (w/o Inherent OC, w/o Target Inversion)

1: **Require:** pretrained $u_t^{\theta^*}$, $\rho$
2: $u_t^\theta \leftarrow u_t^{\theta^*}$
3: **for** $x_0 \sim p_{\text{input\_frames}}, x_1 \sim p_{\text{future\_frames}}$ **do**
4: $\quad \mu_1, \sigma_1 = \text{VAE}(x_1), \quad x_1 \leftarrow \mu_1$
5: $\quad \mu_0, \sigma_0 = \text{VAE}(x_0), \quad x_0 \leftarrow \mu_0$
6: $\quad t \sim \mathcal{U}[0, 1]$
7: $\quad x \leftarrow (1 - t)x_0 + tx_1$
8: $\quad \mathcal{L}_{\text{CFM}}(\theta) = \left\| u_t^\theta(x) - (x_1 - x_0) \right\|^2$
9: $\quad$ Update $\theta$ using GD on $\mathcal{L}_{\text{CFM}}(\theta)$
10: **end for**

---

---

**Algorithm 3** Flowing From Observed To Future Frames (w/ Inherent OC, w/o Target Inversion)

1: **Require:** pretrained $u_t^{\theta^*}$, $\Pi(p_{\text{input\_frames}}, p_{\text{future\_frames}})$, $\rho$
2: $u_t^\theta \leftarrow u_t^{\theta^*}$
3: **for** $x_0, x_1 \sim \Pi$ **with** $(x_0, x_1)$ **inherent optimal couplings do**
4: $\quad \mu_1, \sigma_1 = \text{VAE}(x_1), \quad x_1 \leftarrow \mu_1$
5: $\quad \mu_0, \sigma_0 = \text{VAE}(x_0), \quad x_0 \leftarrow \mu_0$
6: $\quad t \sim \mathcal{U}[0, 1]$
7: $\quad x \leftarrow (1 - t)x_0 + tx_1$
8: $\quad \mathcal{L}_{\text{CFM}}(\theta) = \left\| u_t^\theta(x) - (x_1 - x_0) \right\|^2$
9: $\quad$ Update $\theta$ using GD on $\mathcal{L}_{\text{CFM}}(\theta)$
10: **end for**

---

## C  ADDITIONAL VISUAL RESULTS

This section presents additional qualitative results of predicted videos from FlowFrames (Fig. 7), visual comparisons from ablations on its principal design choices (Fig. 8), as well as studies varying the number of neural function evaluations (NFEs) (Fig. 9) and the number of future frames (Fig. 10). The observed videos are taken from the OpenVid validation set and the predicted videos are generated with our model fine-tuned from LTXV. We set the number of input and future frames to $41$ and resolution to $256 \times 384$ in the examples provided in Fig. 7, Fig. 8 and Fig. 9. All examples are provides in a video format in the supplementary material.

Fig. 7 shows the observed frames provided as input, the corresponding ground-truth future frames, and the predictions generated by our method. FlowFrames produces video continuations that exhibit detailed, physically plausible motion and strong temporal coherence with the observed context; all achieved without explicit conditioning mechanisms, but simply by taking the observed frames directly as input. For instance, in the Qualcomm example, the camera's rotation around its center in the ground truth is mirrored with fidelity in the predicted frames. Similarly, the motion of the boat traversing left to right with accompanying water dynamics is reproduced with coherent detail. In another case, the car advancing from the background toward the foreground continues seamlessly in our predictions, maintaining consistency with both camera dynamics and temporal flow. Thus, FlowFrames preserves logical structure and temporal realism in video prediction.

Fig. 8 presents the observed frames, the ground-truth future frames, and the results produced by four variants: training from scratch with Inherent OC and TI, fine-tuning from LTXV without OC and TI, fine-tuning with OC but without TI, and fine-tuning with both OC and TI. Training from scratch with OC and TI, leads to blurring and visual artifacts, reflecting poor convergence and the significance of using a pre-trained model as an initialization. Similarly, fine-tuning without OC and TI produces unstable results, with noticeable artifacts and color discrepancies (e.g., in the boat and car examples). By contrast, incorporating both OC and TI during fine-tuning yields markedly improved visual fidelity. For instance, in the sea-and-mountain example, the inherent OC-only setting generates blurred sand textures, while the joint inherent OC and TI setting restores sharper, more realistic details, as highlighted by the red boxes in Fig. 8.

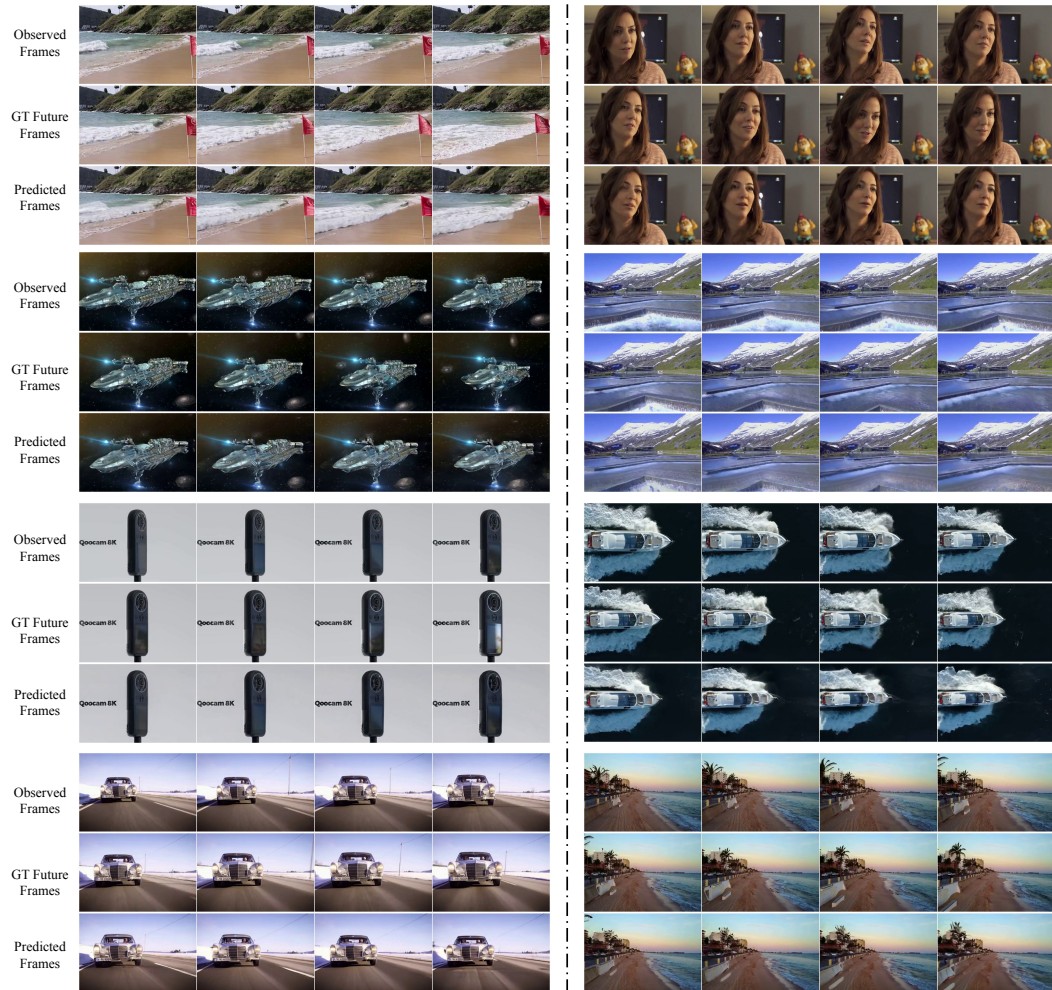

Figure 7: Additional visual results on the OpenVid validation set. FlowFrames, fine-tuned from LTXV, generates video continuations that are both temporally coherent and physically realistic. Frames are visualized with a stride of 13. GT stands for ground truth. For an enhanced viewing experience, please refer to the accompanying videos provided in the supplementary material.

Fig. 9 illustrates the observed frames, the ground-truth future frames, and the videos generated by our method with 1, 5, 6, 10, and 40 NFEs. As shown, a single NFE is inadequate for video prediction, yielding results with severe motion blur (e.g., in the white car sequence). In contrast, using 5–10 NFEs produces video predictions of competitive fidelity to that obtained with 40 NFEs, demonstrating that our method achieves good visual quality with reduced NFEs.

Fig. 10 depict the observed frames, the ground-truth future frames and generated video results of long video prediction. In this example, the number of observed and future frames is set to 113. Although FlowFrames is trained only on 17- and 41-frame video chunks, it generalizes to substantially longer predictions. In the sea sequence, the generated flow of water from left to right closely follows the ground truth. In the ice-over-water sequence, forward camera motion is faithfully preserved, yielding predictions that remain temporally coherent with the observed frames.

## D LIMITATION AND FUTURE WORK

FlowFrames delivers a $2\times$ reduction in input dimensionality and achieves state-of-the-art video prediction in quantitative metrics with substantially fewer NFEs, yet important limitations remain.

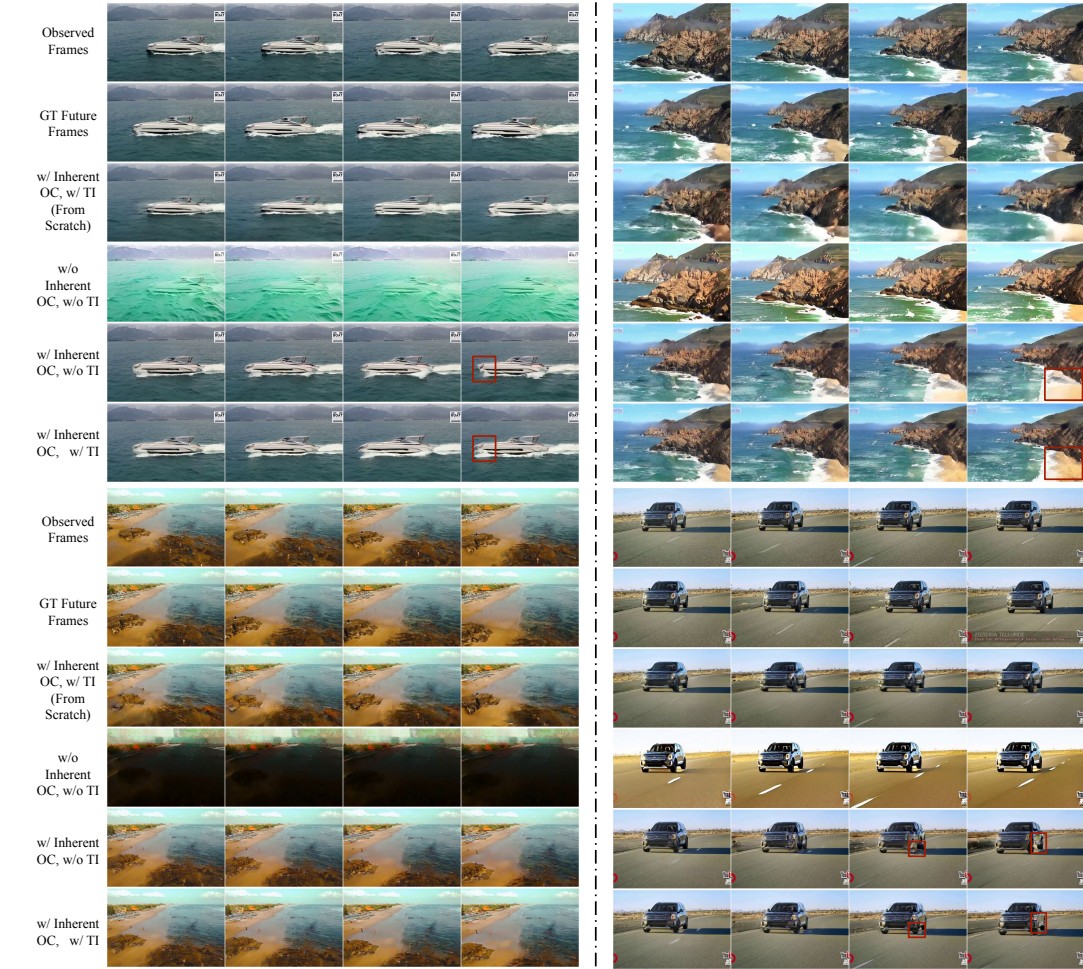

Figure 8: Additional visual results on ablation across four training setups (frames shown with stride 13). Training from scratch with OC+TI produces visual artifacts. Fine-tuning without OC+TI results in predicted videos with temporal inconsistencies with the observed frames and color shifts. Incorporating both OC and TI during fine-tuning yields sharper details than OC-only fine-tuning (see red boxes). GT refers to ground truth. Videos are included in the supplementary material for improved visualization.

Below, we elaborate on the constraints outlined in the main text and describe corresponding avenues for future work.

**Heterogeneous Video Chunk Lengths.** In the main text, we showed that FlowFrames maintains stable FID/FVD and strong visual quality for long video prediction with visuals in Fig. 9 with 113 observed and future frames; despite training solely on 17- and 41-frame chunks. Beyond this range, however, results become degraded with visible interpolation artifacts with the observed frames. Fig. 11 illiterates representative failure cases at longer video prediction (129 observed and future frames). Consistent with prior evidence in text-to-image (Podell et al., 2023; Esser et al., 2024; Dai et al., 2023; Xie et al., 2024) and text-to-video (Zheng et al., 2024; Peng et al., 2025; Yang et al., 2025; Wan et al., 2025; Kong et al., 2025; HaCohen et al., 2024; Zhou et al., 2024; Chen et al., 2025b; Seawead et al., 2025) methods, exposure to diverse spatial and temporal scales during training improves inference-time robustness. A natural next step is therefore to train the proposed method with heterogeneous video chunk lengths, broadening temporal coverage and strengthening generalization for very long predictions.

**Controllability.** While FlowFrames introduces a new perspective on video prediction; flowing directly from observed to future frames; yielding both a twofold reduction in input dimensionality

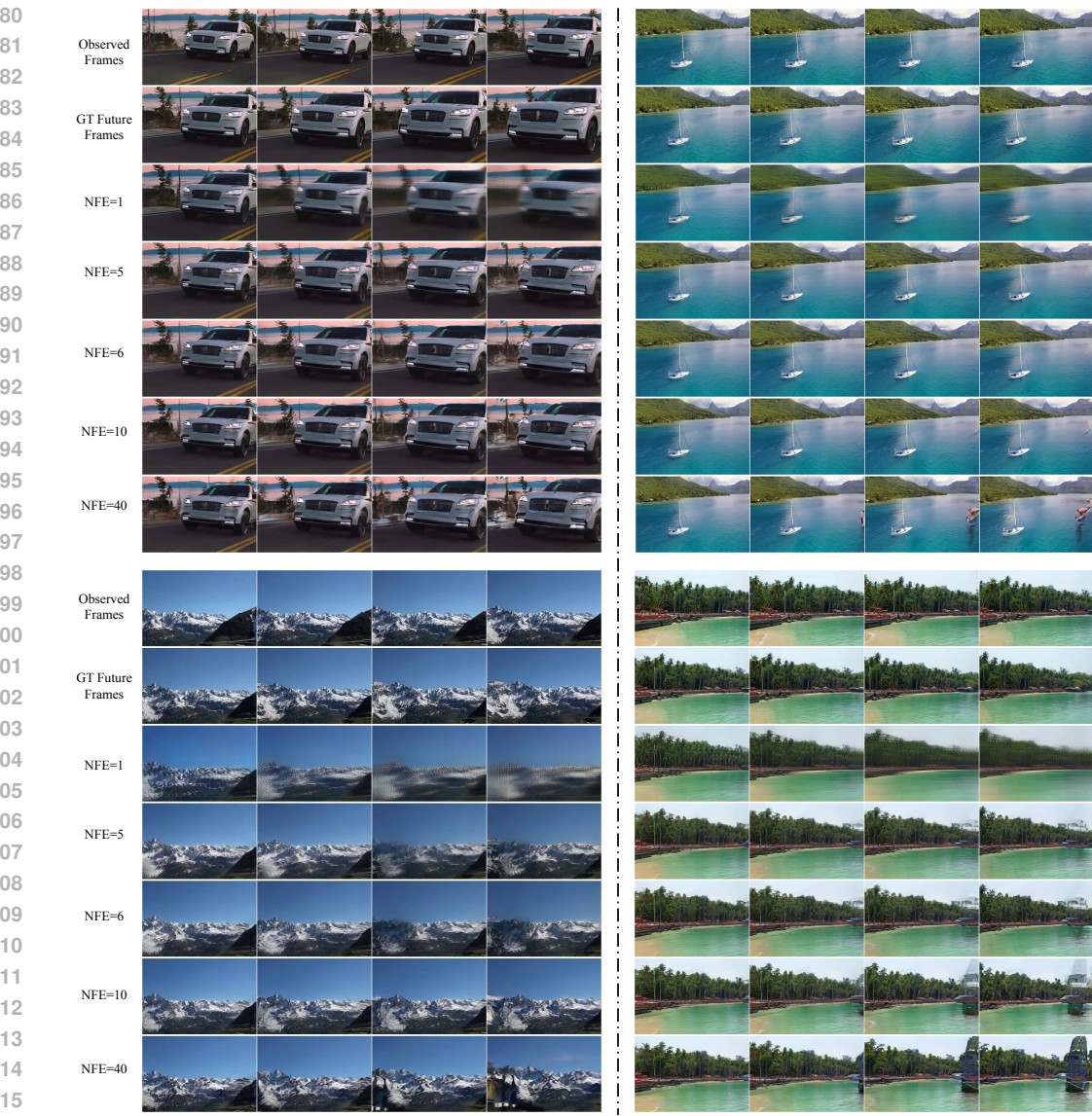

Figure 9: Qualitative results for the ablation on neural function evaluations (NFEs). Frames are shown with a stride of 13. Using 5–10 NFEs yields video predictions with fidelity comparable to 40 NFEs, while using NFE=1 results in blurred video outputs. GT abbreviates ground truth. For a more comprehensive assessment, please refer to the videos in the supplementary material.

and fewer neural function evaluations compared to existing methods, the current framework does not leverage additional conditioning signals such as text prompts, depth maps, motion or camera trajectories. After the arrival of diffusion (Sohl-Dickstein et al., 2015; Song et al., 2021a; Ho et al., 2020; Song et al., 2021b; Dhariwal & Nichol, 2021; Liu et al., 2022; Lipman et al., 2023) and flow (Lipman et al., 2023; Liu et al., 2022; Albergo & Vanden-Eijnden, 2023) models controllability has become a key in image and video generation (Zhang et al., 2023a;b; Jiang et al., 2025; He et al., 2025; Lei et al., 2025; Xiao et al., 2024; Xu et al., 2024; Liang et al., 2024; Zheng et al., 2025; Hou & Chen, 2025; Ma et al., 2025c; Zhou et al., 2025). Extending FlowFrames to incorporate such auxiliary signals is therefore a promising direction for future work, enabling richer, more adaptable, and user-guided video prediction.

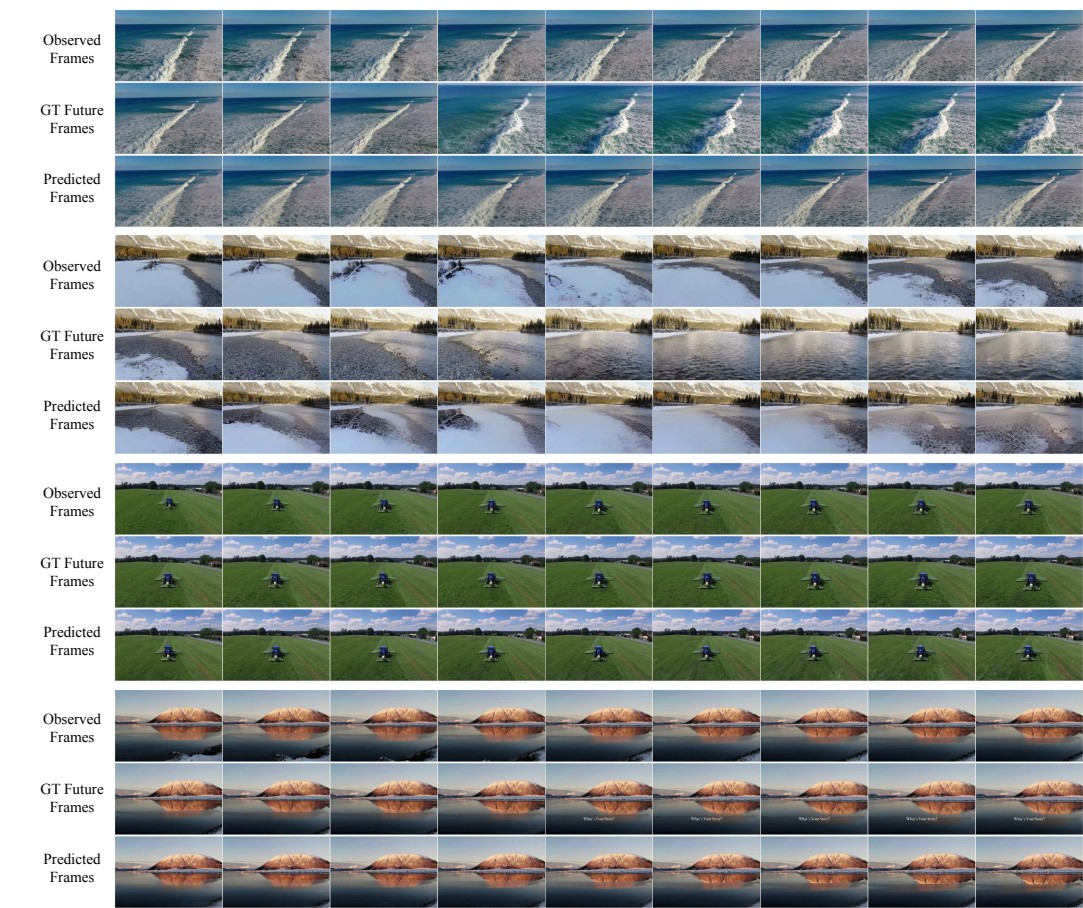

Figure 10: Visual results for the ablation on long video prediction. The number of observed and future frames is 113 and the frames are visualized with a stride of 14. Despite being trained only on 17- and 41-frame sequences, FlowFrames successfully generates coherent long-video predictions. GT indicates ground truth. The supplementary material provides video examples for improved interpretation of these results.

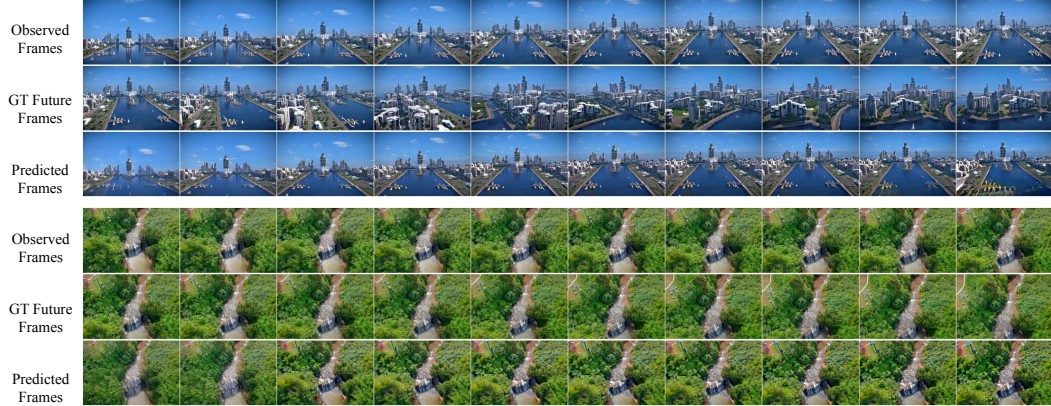

Figure 11: Failure cases for very long prediction. Shown are 129 observed and future frames (visualized with a stride of 14). Beyond 113-frame video chunks, FlowFrames degrades, exhibiting interpolation artifacts and reduced coherence with the observed context. Videos are included in the supplementary material for improved visualization.

