# OpenReview forum: "Flowing From Observed To Future Frames For Efficient Video Prediction"
_ICLR.cc/2026/Conference — ICLR 2026 Conference Withdrawn Submission_

### Official Review · Reviewer_XPKX · 2025-10-31

**Soundness:** 2
**Presentation:** 3
**Contribution:** 3
**Rating:** 4
**Confidence:** 3

**Summary:**

The authors propose a new method for fast and efficient video prediction. The core idea is to adapt a pre-trained T2V model to learn a direct mapping from the distribution of observed video frames to the distribution of future video frames, rather than the conventional approach of conditioning on observed frames and generating from noise. The proposed method achieves competitive results in terms of both quality and efficiency.

**Strengths:**

- The authors explored a novel method for generative video prediction, which directly learn a vector field from the observed frames to the future frames.
- The proposed method has achieved competitive quantitative and qualitative results on the provided benchmarks.

**Weaknesses:**

- The qualitative comparisons are conducted on relatively simple scenarios without complex motion, occlusions, or scene changes. This makes it difficult to fully assess the predictive capability of the proposed method.
- The paper lacks comparisons with traditional, non-generative approaches. While generative models are powerful in handling large and complex motions, they may not perform as effectively or efficiently as traditional video prediction methods in simpler motion settings. Including such baselines would provide a more comprehensive and insightful evaluation of the proposed method’s contribution and performance.

**Questions:**

- Could the authors include more qualitative results for the difficult cases of complex motion, occlusions, or scene changes?
- Is it possible to provide additional results on other benchmarks, such as DAVIS[1] and KITTI[2]?
- It is necessary to include the non-generative method, e.g., DMVFN[3], for comparative results for comprehensive evaluation.

[1] The 2017 davis challenge on video object segmentation.

[2] Vision meets robotics: The kitti dataset.

[3] A Dynamic Multi-Scale Voxel Flow Network for Video Prediction.

---

### Official Review · Reviewer_WzSC · 2025-10-31

**Soundness:** 3
**Presentation:** 3
**Contribution:** 2
**Rating:** 4
**Confidence:** 5

**Summary:**

The paper proposes FlowFrames, a methodology for training video flow models to learn the flow between past and future frames in contrast to the conventional way of going from noise to future frames conditioned on the past.

**Strengths:**

The idea is interesting and the method is simple enough. The memory usage and sampling efficiency gains are significant.

**Weaknesses:**

1. The main concern is about the apparent common failure case, where the predicted video chunk looks similar to the observed one (e.g. videos in Figure 10 and 11). This is actually an expected outcome of the flow matching framework. Flow matching leads to approximating OT between the source and the target distributions. If a chunk is both in the source and the target distributions, in some cases flow matching can couple it to itself, resulting in near-zero flow field around that chunk. When the dataset size grows the past and the future chunk distributions converge to each other and the chosen Inherent Optimal Coupling may even hurt the training. Therefore, the outcome of the training may largely depend on the diversity and the scale of the data. This questions the overall motivation of flowing from past to future frames. Could the authors provide some discussion on this and maybe some observations on how common this failure case is?
2. The idea of learning the flow field between the past and the future observations was previously studied in [1]. Although the application to fine-tuning video models requires to solve domain-specific challenges, the discussion on [1] would better justify the design choices made in the paper.
3. The idea of Target Inversion is interesting, but needs more discussion/analysis. E.g. what would be the effect of varying $\sigma_0$?

[1] Lim, Soon Hoe, et al. "Elucidating the design choice of probability paths in flow matching for forecasting." arXiv preprint arXiv:2410.03229 (2024).

**Questions:**

Missing definitions:

1. $z$ in Equation (1) was never defined.
2. Coupling was not defined.

Other questions:
1. In Figure 4, it appears that after a few training steps the blue model is the best among couterparts (even compared to fully trained models), which is counterintutive. Could the authors explain this? Besides this, the plots are missing the base model's FID and FVD.
2. How is FVD computed in Figure 6b? Is the same number of frames used to calculate FVD for different number of future frames?

---

### Official Review · Reviewer_vLPH · 2025-11-01

**Soundness:** 3
**Presentation:** 3
**Contribution:** 2
**Rating:** 2
**Confidence:** 5

**Summary:**

This paper, Flowing from Observed to Future Frames for Efficient Video Prediction, claims to introduce a novel and efficient method for video prediction using deterministic flow matching. The authors fine-tune pretrained text-to-video flow models  to learn a direct mapping from observed frames to future frames, eliminating Gaussian noise and reducing input dimensionality. While the implementation is neat and the reported memory/NFE savings are real, the paper fundamentally misunderstands its own task and overstates its novelty.

**Strengths:**

**Empirical efficiency gains.**

The claimed improvements in GPU memory and computational cost are significant and well-documented, providing a tangible engineering benefit even if conceptual novelty is limited.

**Simple yet effective design.**
Removing stochastic noise and simplifying the input space yields faster inference without complicated new modules. The method is clean and avoids heavy architectural overhead.

**Weaknesses:**

This paper does not perform “video prediction” as defined in the forecasting community. It performs deterministic video continuation under a generative evaluation setting. That conceptual confusion undermines the entire paper.

**Task–Metric Mismatch**
FlowFrames is fully deterministic—flow matching is an ODE, not an SDE. Given an input chunk, the output is uniquely determined. There is no stochasticity, uncertainty modeling, or multimodality.

Yet, all evaluations use generative realism metrics (FID/FVD) on OpenVid and NuScenes. These metrics measure perceptual diversity and realism, not predictive accuracy. For a deterministic predictor, such metrics are meaningless. A proper evaluation would use MSE, PSNR, SSIM, or LPIPS on standard prediction datasets.

**Wrong Datasets, Wrong Baselines**

If the model truly targets video prediction, it must be tested on forecasting benchmarks such as Moving MNIST, Caltech Pedestrian, KTH Human Actions, WeatherBench, or SEVIR, and compared with established deterministic forecasting models (SimVP, PredRNN++, PhyDNet, EarthFormer, PreDiff).
Instead, the authors only compare against generative T2V or world models (Vista, GEM, LTXVCondition, CausVid) — models designed for perceptual generation, not accurate prediction.

**Evaluation Inconsistency**
A deterministic model pretending to be a stochastic generator is nonsensical. FlowFrames can never produce diverse outcomes for an ambiguous future, so it cannot be fairly compared to stochastic generative models. The “state-of-the-art” FID/FVD claims are irrelevant to prediction quality.

**Questions:**

1. Since flow matching is inherently deterministic, how can FlowFrames represent stochastic uncertainty in future prediction?

2. Why were predictive datasets (e.g., Moving MNIST, WeatherBench) and accuracy metrics (MSE, PSNR, SSIM) not included?

---

### Note · Authors · 2025-11-14

I have read and agree with the venue's withdrawal policy on behalf of myself and my co-authors.